# Targeting the Unfolded Protein Response as a Disease-Modifying Pathway in Dementia

**DOI:** 10.3390/ijms23042021

**Published:** 2022-02-11

**Authors:** Emad Sidhom, John T. O’Brien, Adrian J. Butcher, Heather L. Smith, Giovanna R. Mallucci, Benjamin R. Underwood

**Affiliations:** 1Department of Clinical Neurosciences, University of Cambridge, Cambridge CB2 0AH, UK; es839@medschl.cam.ac.uk (E.S.); ajb342@medschl.cam.ac.uk (A.J.B.); hls59@medschl.cam.ac.uk (H.L.S.); gm522@medschl.cam.ac.uk (G.R.M.); 2Department of Clinical Neurosciences, UK Dementia Research Institute, University of Cambridge, Cambridge CB2 0AH, UK; 3Cambridgeshire and Peterborough NHS Foundation Trust, Windsor Research Unit, Fulbourn Hospital, Cambridge CB21 5EF, UK; 4Gnodde Goldman Sachs Translational Neuroscience Unit, Windsor Research Unit, University of Cambridge, Cambridge CB2 1TN, UK; 5Department of Psychiatry, University of Cambridge, Herchel Smith Building, Forvie Site, Cambridge CB2 0SZ, UK; john.obrien@medschl.cam.ac.uk

**Keywords:** dementia, Alzheimer’s disease, trazodone, unfolded protein response, integrated stress response, neuroprotection, neurodegenerative disorders, neurocognitive disorders

## Abstract

Dementia is a global medical and societal challenge; it has devastating personal, social and economic costs, which will increase rapidly as the world’s population ages. Despite this, there are no disease-modifying treatments for dementia; current therapy modestly improves symptoms but does not change the outcome. Therefore, new treatments are urgently needed—particularly any that can slow down the disease’s progression. Many of the neurodegenerative diseases that lead to dementia are characterised by common pathological responses to abnormal protein production and misfolding in brain cells, raising the possibility of the broad application of therapeutics that target these common processes. The unfolded protein response (UPR) is one such mechanism. The UPR is a highly conserved cellular stress response to abnormal protein folding and is widely dysregulated in neurodegenerative diseases. In this review, we describe the basic machinery of the UPR, as well as the evidence for its overactivation and pathogenicity in dementia, and for the marked neuroprotective effects of its therapeutic manipulation in murine models of these disorders. We discuss drugs identified as potential UPR-modifying therapeutic agents—in particular the licensed antidepressant trazodone—and we review epidemiological and trial data from their use in human populations. Finally, we explore future directions for investigating the potential benefit of using trazodone or similar UPR-modulating compounds for disease modification in patients with dementia.

## 1. Introduction

Disease-modifying drugs for neurodegenerative diseases leading to dementia are urgently needed. In England and Wales, the annual mortality from dementia is similar in magnitude to the mortality arising from SARS-CoV-2 in 2020, at nearly 70,000 deaths per year—and greater than that from ischemic heart disease or cerebrovascular disease [1,2]. The biggest risk factor for developing dementia is advancing age [3]. As the population ages, the number of cases will double every 20 years, with the majority of cases in developing countries [4]. The most common cause of dementia is Alzheimer’s disease (AD), which accounts for 60–70% of all cases [5]. The current drug treatments available—such as cholinesterase inhibitors developed after initial trials of tacrine and the NMDA receptor antagonist memantine—may ameliorate the symptoms of the disease, but they do not delay its progression [6,7]. Even a modest delay in disease progression is likely to have significant benefits; slowing the onset of AD by five years has been predicted to decrease disease incidence and associated costs by ~40% [8]. This would be significant at the medical and societal levels, reducing personal suffering, but also economically: dementia care costs some GBP 26 billion per year in the UK alone, which is anticipated to double in the next 25 years without disease-modifying treatments [9].

The many mechanisms driving the pathology of Alzheimer’s disease are becoming increasingly understood; they include disease-specific processes—due to β-amyloid deposition, for example—as well as more generic common processes, resulting from homeostatic responses to protein aggregation, such as the unfolded protein response (UPR), dysregulated autophagy, inflammation, and oxidative stress [10,11,12,13]. These common mechanisms are gathering attention as therapeutic targets [14] due to their broad relevance across neurodegenerative diseases. Drugs in clinical use for other conditions that target these pathways include, for example, the autophagy inducer rilmenidine [15], and trazodone [16], which acts on the UPR. These are therefore prime candidates for repurposing. This review describes the cellular mechanisms and preclinical studies underpinning therapeutic approaches focusing on UPR modulation and describes evidence from epidemiology and early clinical trials using trazodone—for reasons other than disease modification—before exploring how these could be turned into available treatments for patients with dementia.

## 2. The Unfolded Protein Response

The UPR is a highly conserved cellular stress response that acts to restore protein homeostasis upon accumulation of misfolded proteins in the endoplasmic reticulum (Figure 1 and Figure 2). UPR activation results in decreased rates of protein translation, induction of chaperones, and increased ER folding capacity. These outcomes are mediated by the activation of three ER-resident transmembrane proteins that sense and respond to ER stress: activating transcription factor 6 (ATF6), the inositol-requiring enzyme 1 (IRE1), and protein kinase RNA-like endoplasmic reticulum kinase (PERK) [17]. Under resting conditions, the three sensors are held in an inactive state through their association with binding immunoglobulin protein (BiP) in the ER lumen. Under stress conditions, BiP dissociates, preferentially binding to misfolding proteins, leading to sensor activation and initiation of the UPR signalling cascade [18]. Activated PERK, IRE1, and ATF6 signal through the transcription factors ATF4, XBP1s, and ATF6N, respectively, to drive a coordinated program of transcriptional changes, up-regulating chaperones and components of the ER-associated degradation pathway. If the source of stress cannot be resolved and UPR signalling is prolonged, ATF4 upregulates pro-apoptotic C/EBP homologous protein (CHOP), initiating the apoptotic cascade [19].

Amongst the three branches of the UPR, the PERK branch is unique in that it also drives a translational response to ER stress. Activated PERK phosphorylates the α-subunit of eukaryotic initiation factor 2 (eIF2) [20], which lies at the hub of the related integrated stress response (ISR) of which the PERK branch is a key signalling pathway [21] (see Figure 2). This phosphorylation event switches eIF2 from a substrate to a non-competitive inhibitor of eIF2B—the guanine exchange factor that recycles eIF2-GDP to eIF2-GTP [22]. In the GDP-bound state, eIF2 is unable to form the ternary complex with initiator methionine tRNA; thus, translation is inhibited at the level of initiation. This inhibition is beneficial in the short term, lowering the entry of newly synthesised proteins into the ER and reducing any additional folding burden on the organelle. The repression of translation, however, is transient, achieved via a negative feedback loop whereby phosphorylated eIF2α drives the non-canonical expression of ATF4 which, in turn, upregulates GADD34—the regulatory subunit of the protein phosphatase 1 complex that dephosphorylates eIF2α [23]. Dysregulation of this feedback loop, and the resultant sustained repression of protein synthesis, is a key neurotoxic mechanism in neurodegenerative diseases, as the production of proteins essential for neuronal health and synaptic integrity is impaired [24,25,26].

## 3. Evidence for UPR as a Key Pathogenic Mechanism in Neurodegeneration

Most cases of progressive neurodegenerative diseases are characterised by the misfolding of proteins, which aggregate into intraneuronal proteinaceous inclusions, such as Aβ and tau in Alzheimer’s disease and α-synuclein in Parkinson’s disease. Though most of these proteins are intracytoplasmic, it is important to note that Aβ, though processed in the endoplasmic reticulum and Golgi, accumulates primarily in extracellular plaques, although intraneuronal accumulation has also been observed [27,28].

Postmortem studies in patients with Alzheimer’s disease, Parkinson’s disease, progressive supranuclear palsy, and ALS [29,30,31] reveal markers of UPR activation, including phosphorylation of PERK and eIF2α associated with protein aggregates [32,33] (Figure 1). Importantly, in Alzheimer’s disease, this has been shown to be a feature of early disease and is likely to precede tau tangle formation and paralleling misfolded protein deposition as it spreads through the brain, which makes it a promising target for therapeutic intervention [34]. Downstream mediators of cell death, such as CHOP, have also been demonstrated in postmortem samples from human sporadic ALS patients [31].

**Figure 1 ijms-23-02021-f001:**
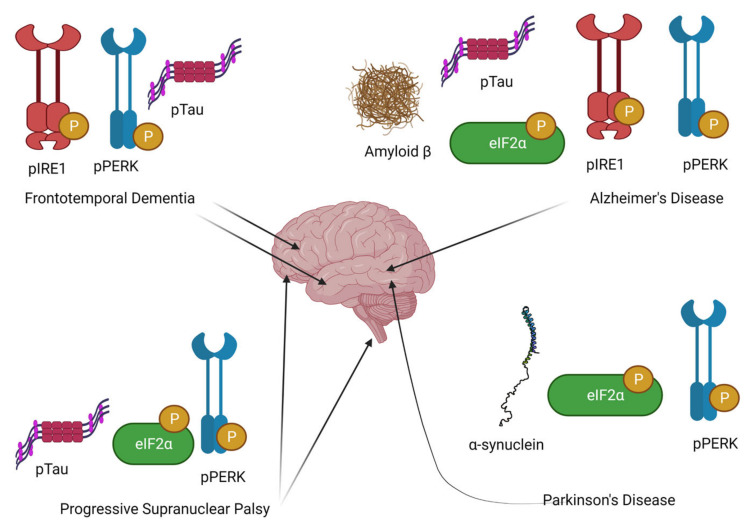
Markers of UPR activation associated with pathological protein deposition in neurodegenerative disordersProteins associated with UPR activation are seen in association with protein aggregates in specific brain regions in patients with different neurodegenerative diseases. Inositol-requiring enzyme 1 (IRE1) and protein kinase RNA-like endoplasmic reticulum kinase (PERK) are key sensory proteins for ER stress, and are activated in frontotemporal dementia and Alzheimer’s disease. p-PERK and its downstream substrate eukaryotic initiation factor 2 (eIF2)—p-eIF2α—are also activated in Parkinson’s disease, Alzheimer’s disease, and progressive supranuclear palsy. For a review, see Hetz and Saxena (2017) [35].

**Figure 2 ijms-23-02021-f002:**
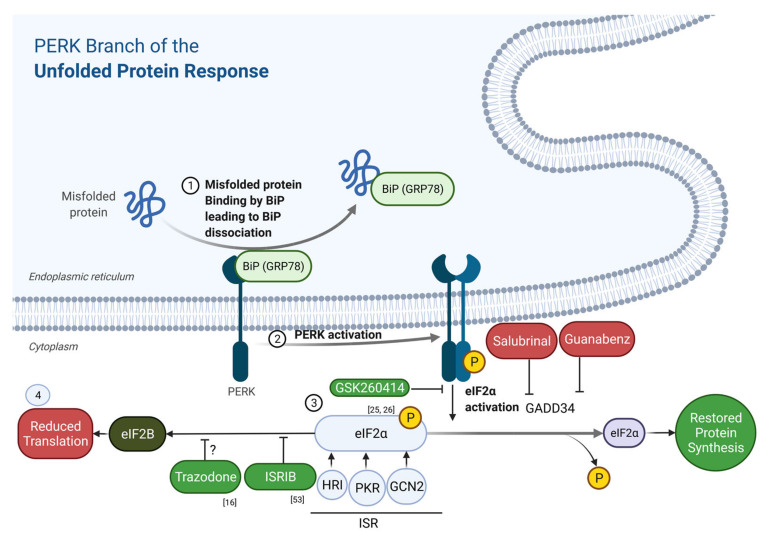
Schematic showing points of action of drugs and experimental compounds targeting the PERK branch of the UPR and ISR. (1) The chaperone protein known as binding immunoglobulin protein (BiP) binds to misfolded proteins, resulting in (2) phosphorylation of PERK. PERK activation, in turn, leads to (3) phosphorylation of eIF2α, which reduces protein translation (4) via by preventing nucleotide exchange by eIF2B needed for initiation of translation. GSK260414 inhibits PERK phosphorylation; ISRIB prevents eIF2B binding by p-eIF2α, and trazodone also acts downstream of p-eIF2α by restoring global protein synthesis rates. Intervention at all of these steps provides exceptional neuroprotection in many animal models of neurodegenerative diseases.

Murine models of neurodegenerative diseases—including prion disease, tauopathies, Alzheimer’s disease, and motor neuron disease—similarly exhibit biochemical markers of UPR overactivation, including sustained levels of phosphorylated PERK and eIF2α from early disease [36,37,38,39]. The importance of the UPR in pathogenesis is shown by the remarkable clinical improvements following genetic or pharmacological inhibition of the PERK-eIF2α pathway, due to partial restoration of global protein translation rates [24,25,26,40,41,42] these include rescue of memory, reversal of behavioural and motor impairments, prevention of neuronal loss, and increased survival in many murine models. The restoration of protein synthesis rates is central to neuroprotection, providing essential synaptic and other proteins necessary for memory formation and neuronal survival. In prion-diseased mice, genetic reduction of eIF2α-P levels through overexpression of GADD34—both in neurons [24] and in astrocytes [43]—is profoundly neuroprotective, restoring synapse numbers, synaptic transmission, and memory, and preventing neuronal loss. In the context of Alzheimer’s disease models, further indirect mechanisms may be involved following UPR activation—for example, via PERK activation of glycogen synthase kinase 3β, and subsequent increased tau phosphorylation and beta amyloid precursor cleaving protein (BACE) expression, increasing amyloidogenesis [44]. Restoring protein synthesis through modulation of eIF2α-P levels by manipulating other ISR kinases is also neuroprotective, and boosts cognition [45,46,47]. For example, genetic modulation of signalling through PERK and another ISR eIF2α kinase—GCN2—reduces levels of eIF2α-P, and is protective in murine models of Alzheimer’s disease, restoring memory and cognitive function [39]. In Down syndrome mice, genetic modulation of PKR signalling restores memory and corrects translatome deficits [40]. 

## 4. Drugs Targeting UPR and ISR

The PERK inhibitor GSK2606414 is profoundly neuroprotective in murine models of prion disease [41], and has shown improvement in murine models of Parkinson’s disease [48] and FTD [26] and fly models of amyotrophic lateral sclerosis (ALS) [42], while salubrinal—an inhibitor of eIF2α dephosphorylation—has the opposite effect [24] (see Figure 2). The clinical ‘cure’ seen in these animal models is associated with decreased synapse and neuronal loss, mediated by restoring protein synthesis rates. However, the clinical utility of GSK2606414 is limited by its systemic (pancreatic) toxicity, precluding its use in humans. The small molecule ISRIB (integrated stress response inhibitor B) acts downstream of phosphorylated eIF2α, preventing its interaction with its target eIF2B, and restores neuronal protein synthesis rates without toxic effects on the pancreas [49,50]. ISRIB boosts memory in wild-type mice, and is neuroprotective and/or enhances cognition in a variety of murine models of neurodegenerative diseases [50,51,52,53]; it has been shown to increase hippocampal protein production and rescue disease phenotypes in murine models of Alzheimer’s disease, but its low level of solubility limits its applicability for human use [54]. To find safe alternatives with similar efficacy in UPR inhibition as ISRIB, Halliday et al. screened an NINDS library of drugs that largely have FDA approval and are known to be neurologically active; they identified dibenzoylmethane (DBM) and the licensed antidepressant trazodone as compounds able to cross the blood–brain barrier, restore protein synthesis rates, and profoundly improve memory and survival in prion-diseased mice and a murine model of frontotemporal dementia by acting downstream of eIF2α phosphorylation, similar to ISRIB [16]. Trazodone is a phenyl piperazine derivative that has a range of agonist and antagonist activity, including antagonism at 5HT2 receptors and agonism at 5HT1A receptors, in addition to adrenergic and histaminergic activity and serotonin uptake inhibition; it is a particularly promising drug, as it has a long record of human use, including in neurodegenerative diseases—for example, in sleep disturbance in Alzheimer’s disease [55].

Trazodone may have pleiotropic therapeutic mechanisms, including effects on tau and restoration of slow-wave sleep [56,57]. Other drugs that impact on the UPR have been developed, and some have now entered clinical trials, e.g., guanabenz for amyotrophic lateral sclerosis [58]. Guanabenz, like salubrinal and sephin-1 in preclinical studies [59], reduces ER stress and protein overload through a mechanism opposite to that of trazodone and ISRIB, as it leads to increased levels of phosphorylated eIF2α; this allows increased chaperone expression, and has proven effective in some murine models of ALS and in the rare hereditary neuropathy CMT1b [58,59,60], underlining the importance of the UPR in these disorders, and highlighting the need to understand the complex mechanisms and outcomes of drugs acting on this pathway.

## 5. Epidemiological Data from Observational Studies on Trazodone Use in Dementia

Given the efficacy of trazodone for neuroprotection in animal models and its wide availability in clinical practice, its potential for repurposing for the treatment of dementia appears clear; its widespread clinical use for many years allows examination of the literature to yield valuable information on (1) the safety of use of trazodone and (2) any potential disease-modifying effects, or lack thereof, of trazodone use in dementia. 

Studies looking at safety in large prescription datasets show trazodone to be similar to atypical antipsychotics and benzodiazepines in terms of fall and fracture risk in demented patients, but with a lower overall mortality risk [61,62]. The largest study of cardiac side effects of trazodone in older people found no effect on QTc and minimal effect on pulse or other ECG parameters [63,64]. Similarly, trazodone may be less likely to result in hyponatraemia than other antidepressants [65].

In terms of disease-modifying effects of trazodone in dementia, several recent studies have looked at naturalistic datasets, with largely negative results. An investigation of over 4000 trazodone users matched to patients prescribed with other antidepressants found a higher overall incidence of dementia diagnosis in the trazodone group, though did not suggest a causal association [66]. This study included patients over 50 years of age prescribed trazodone on two or more consecutive occasions at any time over 17 years; around one-third of patients had a diagnosis of depression. A second study examined routinely collected data from 406 patients with a diagnosis of dementia who were prescribed trazodone and followed up for an average of 2.2 years, with cognitive testing every 4–6 months; no positive impact of trazodone on cognition compared to other antidepressants was found in this study either [67]. Two points are worth noting here: first, the patient group had moderately advanced disease on entry to the study (mean baseline Mini-Mental State Examination (MMSE) 13.7–18.1 in those prescribed trazodone across three sites), and moderate/advanced disease may be beyond the point of possible positive impact—preclinical studies all test the drug and related interventions in relatively early stages, rather than advanced disease. Second, the mean dose of trazodone used was 100 mg; the trazodone dose predicted to impact the UPR in humans is calculated to be around 200 mg, using surface area normalisation formulae used to convert from effective doses in mice [16]. A further observational study found an improvement in the Neuropsychiatric Inventory, but not in cognition, in patients with Alzheimer’s disease treated with trazodone over 12 months, though the number of patients was small, including only eight patients on trazodone; their MMSE score at baseline was 20.0 ± 5, and at 12 months was 19.9 ± 5 [68]. While these are very interesting, well-conducted observational studies, they preclude testing the hypothesis of whether trazodone influences cognitive decline in early disease at UPR-effective doses. 

Interestingly, in a targeted observational study, 25 trazodone users with early Alzheimer’s disease (MMSE > 20), mild cognitive impairment, or normal cognition attending a sleep clinic over 4 years were compared to 25 matched controls who did not take the drug. Trazodone users showed notable improvements in sleep and cognition, with significantly slower decline in MMSE over an average duration of four years [57]. Despite the small size of this study, it does suggest that refining the target patient population to early/mild Alzheimer’s disease—or even MCI—combined with longer follow-up in randomised controlled clinical trials, would better elucidate any effect—or lack thereof—of trazodone on the progression of cognitive decline in early dementia.

Overall, observational studies can only ever show associations, and have many confounding variables. For example, trazodone is an unusual choice of antidepressant for older people [66], and it may be prescribed preferentially in those with symptoms suggestive of incipient dementia (e.g., sleep disturbance [69]), or in established dementia for those with behavioural symptoms that are associated with more advanced disease [67,68,69,70], both of which could mask any benefits of the drug. In a recent study of over 50 years of age, prescribed antidepressants including 465,628 patients, only 4596 (<1%) were prescribed trazodone as first-line treatment, suggesting that it is not a common therapeutic choice [66]. In another study, the most common reason for starting trazodone was agitation, which is more common later in the course of the disease [67].

In summary, the literature from observational studies of trazodone’s effects in dementia provides a conflicting picture, with limited support to date for any disease-modifying effect of trazodone. This is perhaps not surprising, as these studies do not report data designed to test the efficacy of trazodone in this context. To properly test the effects of an intervention and establish causation, the only definitive study design is a randomised controlled trial.

## 6. Clinical Trial Data on Trazodone Use in Older Adults, including Individuals with Dementia

Trazodone has been trialled extensively since the 1970s for efficacy in major depressive disorder [71] (including geriatric and post-stroke populations [72]), pain [73], sleep [74], erectile dysfunction [75], akathisia [76], migraine [77], adjustment disorder [78], bulimia [79], oesophageal contraction abnormalities [80], OCD [81], generalized anxiety [82], and alcohol withdrawal [83,84]. The data from these trials provide important safety data on trazodone use. 

In dementia, clinical trials of trazodone have largely focused on neuropsychiatric symptoms and sleep, not disease modification (trazodone is not licensed for the latter). A trial in which 28 patients with Alzheimer’s disease were treated for agitation with haloperidol or trazodone (mean trazodone dose 218 mg) for 9 weeks found no difference in effect between the two drugs on agitation, although trazodone was better tolerated [85]. In another study, 30 AD patients treated with trazodone or placebo for two weeks showed improvement in sleep with trazodone [86], and in a further small trial, trazodone improved Neuropsychiatric Inventory (NPI) scores in frontotemporal dementia [85].

The largest trial of trazodone on behaviour in the context of dementia involved randomisation of 150 Alzheimer’s patients to either haloperidol, trazodone (mean dose 200 mg; range 50–300 mg), behavioural intervention, or placebo for 16 weeks, and found no impact of trazodone on agitation [87]. Crucially, however, this study did demonstrate the feasibility of treating AD patients with trazodone at a dose of 200 mg with few dropouts, and with a side-effect profile similar to placebo. Indeed, these studies are all encouraging with regards to the tolerability of trazodone in this patient group. However, they were all delivered over a short time period and in moderately to severely advanced disease (for example MMSE < 12 in the largest study [87]), and often at lower doses than those thought to impact the UPR. Two small studies—in 13 Alzheimer’s patients and 26 frontotemporal dementia patients—also measured cognition as an outcome of trazodone treatment; no change in Mini-Mental State Examination (MMSE) was found in either of these studies [86,87,88] after 10 and 12 weeks of treatment, respectively. The largest study to date, including 37 patients taking trazodone, found trazodone to have a worse cognitive outcome as measured by the MMSE compared to behavioural management for agitation in Alzheimer’s disease, but trazodone was not significantly different on this measure compared to placebo after 16 weeks of treatment [87]. Both clinical trials and observational studies on the effect of trazodone on cognition in healthy controls and patients with a variety of conditions, including dementia, have been the subject of a recent systematic review that described 4 studies demonstrating impaired cognition with trazodone, 7 showing no effect, and 5 showing cognitive improvement, although the longest treatment period included was 16 weeks for interventional studies [89].

Taken together, clinical trial data thus far support the tolerability of trazodone in patients suffering from dementia, including at predicted potentially UPR-modifying doses of 200 mg [16]. In no study to date has trazodone been used in early-stage disease over a sufficient number of months/years and at the correct dose to draw any conclusions about its potential for disease modification. Most published trials of trazodone in dementia used durations of treatment measured in weeks, and none has met the requirements of the consensus statements for clinical trials of disease-modifying drugs in dementia, which stipulate a minimum trial duration of 18 months [90]. New studies are underway, including in motor neuron disease (MND/ALS), where trazodone was chosen as one of the first two drugs to be entered into a multicentre UK adaptive trial (MND-SMART) [91] to identify disease-modifying drugs in this condition, based on evidence from the most promising neuroprotective compounds in murine models of ALS.

## 7. Future Directions

In summary, overactivation of the UPR—and in particular the PERK/eIF2α pathway—appears to be implicated in the pathogenesis of Alzheimer’s disease and related disorders in human patients, and there is compelling evidence for its role in mediating neurodegeneration in preclinical studies. Furthermore, the profound neuroprotective effects of UPR modulation in multiple murine models of disease, along with the efficacy in boosting cognition in aged and wild-type mice, have led to a major drive for drug discovery in this pathway. The licensed drug trazodone acts on the pathway in question, and could be rapidly repurposed, bypassing years of drug discovery and early-phase clinical trials. Observational studies and clinical trials in patient populations taking trazodone do not clearly signal a protective or disease-modifying effect, but these are beset with confounding factors and are ill-placed to identify disease-modifying effects, due to the populations studied, durations of treatment, dosages, and outcomes used. Ultimately, only a randomised, placebo-controlled clinical trial of trazodone or similar compounds in patients will answer the question as to whether this approach brings relevant clinical benefits. Given the extensive preclinical data identifying the UPR as a therapeutic target in neurodegenerative disease, the efficacy of trazodone in many murine models, and its long track record of safe use in humans, trazodone should now be tested in experimental medicine studies on human patients. Key mechanistic questions include whether impaired cerebral protein synthesis rates—as occur in murine models—can be demonstrated in patient brains using in vivo readouts, at what stage in the disease such a decline (if present) can be detected, whether trazodone can restore cerebral protein synthesis in patients, and whether this protects cognition. With or without such mechanistic data, if trazodone is shown to be successful in clinical trials, it could represent the first disease-modifying drug for dementia, and one that could be rapidly and widely available due to its established nature and low cost—an outcome that would have a major clinical impact worldwide.

## Data Availability

Not applicable.

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
