# Peer review of "Targeting the Unfolded Protein Response as a Disease-Modifying Pathway in Dementia"

_ijms, 2022, doi:10.3390/ijms23042021_

Round 1

Reviewer 1 Report

The manuscript “Targeting the Unfolded Protein Response as a Disease Modifying Pathway in Dementia” provides a review about the importance of unfolded protein responses (UPR) in dementia. It gives an overview about the impact, challenges, and current treatment options of different forms of dementia with an emphasis on Alzheimer`s disease (AD), the proteins and signaling pathways involved in UPR and it discusses the results and possible utilization of experimental drugs.

The first three chapters of the manuscript are well written and show compelling evidence about the potential importance of the regulation of UPR in dementia.

Line 34: “In England and Wales, the annual mortality from dementia is similar in magnitude to the mortality arising from SARS-CoV-2 in 2020 at nearly 70,000 deaths per year” Authors should also compare the mortality of AD to other chronic diseases. Authors should address the age dependent risk of dementia and AD.

Line 51: “common mechanisms (of AD) are gathering attention as therapeutic targets, due in part to the repeated failure of multi-billion-dollar clinical trials targeting disease-specific proteins, notable Aβ, which has diminished confidence in ever finding dementia treatments” It is not a scientific argument, and these kind of personal opinions and conclusions has no place in scientific reviews. The cited article Mullard A (2019) NLRP3 inhibitors stoke anti-inflammatory ambitions. Nat Rev Drug Discov 18:405–408 is a opinion peace / news article. Stating that there is diminished confidence in finding Ab targeting drugs is simply wrong. The US Food and Drug Administration (FDA) just recently approved a drug targeting Ab (aducanumab (Aduhelm)). And there are more coming. (https://www.nature.com/articles/d41586-021-03410-9) Authors should stick to the facts without injecting opinions into their paper.  

Line 109-110. Figure 1 shows the activation of UPR “associated with Pathological Protein deposition in Neuro-109 degenerative Disorders”.
Authors need to increase the clarity of how UPR can affect beta-amyloid (Ab) aggregation in AD. The common understanding is the Ab accumulation occurs in the extracellular space and leads to the formation of senile plaques. The common knowledge about Amyloid precursor protein (APP) is that it is a transmembrane protein located in the cell membrane. The authors are right when they state “Most cases of progressive neurodegenerative diseases are characterized by the misfolding of proteins which aggregate into intraneuronal proteinaceous inclusions such as Aβ and tau in Alzheimer’s disease” but they also should discuss that Ab accumulates outside of the cell. Authors should elaborate on the locations of Ab peptide formation (ER, Golgi, cell membrane) and locations of Ab aggregation (ER, Extra cellular space). The following publications (or similar ones) can be cited to back up these claims: LaFerla, F., Green, K. & Oddo, S. Intracellular amyloid-β in Alzheimer's disease. Nat Rev Neurosci 8, 499–509 (2007). https://doi.org/10.1038/nrn2168. Del Prete et al. J Alzheimers Dis. 2017; 55(4): 1549–1570.

The main problem with the manuscript is in chapter 5, 6 and 7. In these chapters the authors focus on the drug Trazodone and discuss its potential in the treatment of AD. It is unfortunate that the authors did not exercise the expected and required scientific rigor during this discussion.
These chapters include a lot of assumptions about why Trazodone did not work in clinical trials in the past to treat AD. The main issue here is that the authors avoided to give a clear scientific argument about why previous clinical trials were inconclusive but clearly ague that the conclusion of the trials – Trazodone does not treat AD – is wrong.

Line 209 “ This study found no positive impact on cognition compared to treatment with other antidepressants. However, the patient group had relatively advanced disease which may have been past the point of possible positive impact and doses were lower than those proposed to impact the UPR (predicted around 200mg [14]), with a mean of 100mg in this study.”
“Relatively advanced disease” is not a scientific term but a lingo what we use in the cafeteria when we have causal chat with peers and colleagues. Authority should disclose a scientific argument citing data and evidence about what disease severity could be treated with Trazodone indicating the disease severity with currently used dementia ratings.

The insufficient dosage of Trazodone is a very good argument. If previous clinical trials were done with insufficient dose than the outcome of these studies can be deemed inconclusive. Authors should elaborate on what dosage would be sufficient and disclose more relevant details especially because the cited paper (Ref #14 Halliday M, Radford H, Zents KAMM, et al (2017) Repurposed drugs targeting eIF2α-P-mediated translational repression prevent neurodegeneration in mice. Brain 140:1768–1783.) used the following dose: “ Mice were intraperitoneally injected once daily with 40 mg/kg trazodone hydrochloride”. That would be a lot of mg (more than 200) for a 70kg human being. At this point it is unclear why 200mg would be sufficient to treat AD and based on the animal experiment it is unclear what dosage should be sufficient for humans to treat AD. Authors should address this issue and discuss the dosage of Trazodone.

Similar problems arise at line 225-230. Authors disclose opinions and speculations without providing any evidence or facts. Authors should disclose evidence so their opinion can be recognized as a scientific message.

Line 224-226 “Thus, these studies do not show a clear effect of trazodone use on dementia incidence or cognition – with one exception [53], however, observational studies can only ever show associations and have many confounding variables.” Authors should provide more details about that “one exception”. Author should discuss this study and compare it to the other studies which had negative outcome. Is there a difference between them what would help to design future studies to evaluate Trazodone?

Line 227-230 “For example, trazodone is an unusual choice of antidepressant for older people, and it may be prescribed preferentially in those with symptoms suggestive of incipient dementia (for example sleep disturbance) or in established dementia for those with behavioral symptoms which are associated with more advanced disease, both of which would mask any benefits of the drug.” Author must cite relevant literature to back up this claim. Was it prescribed preferentially to a certain patient group or not? Is there any specific evidence based on published and or existing data that the effect of Trazodone could have been masked?

Line 231 “ In summary” Authors disclose opinions and speculations without providing any evidence or facts. Authors should disclose evidence so their opinion can be recognized as a scientific message. What are the specific shortcomings of the cited studies and would be the needed tests to prove the efficacy of Trazodone.
Line 266-268 “Taken together…” Authors, again disclose opinions and speculations without providing any evidence or facts. Authors should disclose evidence so their opinion can be recognized as a scientific message. What would be a long enough treatment and why?

My personal opinion is that I agree with the authors that Trazodone needs a closer look, and it has the potential in the treatment of AD. But my opinion does not matter because science is not about opinions but verified and well-established facts. Every hypothesis must be built on facts. I encourage the authors to provide sufficient evidence and facts to the claims of their review article.

Author Response

Response to reviewers

Targeting the Unfolded Protein Response as a Disease Modifying Pathway in Dementia
Emad Sidhom1,2,3, John O’Brien4, Adrian J Butcher1,2, Heather L Smith1,2, Giovanna Mallucci1,2, Benjamin R Underwood3,4,*

1Department of Clinical Neurosciences, University of Cambridge, Cambridge CB2 0AH: es839@medschl.cam.ac.uk

2   UK Dementia Research Institute at the University of Cambridge, Cambridge CB2 0AH, United Kingdom

3 Cambridgeshire and Peterborough NHS Foundation Trust, Windsor Research Unit, Fulbourn Hospital, Cambridge CB21 5EF

4 Department of Psychiatry, University of Cambridge, Herchel Smith Building, Forvie Site,

Cambridge CB2 0SZ

*   Correspondence: ben.underwood@cpft.nhs.uk

We thank the reviewers for their thorough consideration of our manuscript and valuable suggestions.  We deal with each in turn below with our responses in blue.

Reviewer 1

The manuscript “Targeting the Unfolded Protein Response as a Disease Modifying Pathway in Dementia” provides a review about the importance of unfolded protein responses (UPR) in dementia. It gives an overview about the impact, challenges, and current treatment options of different forms of dementia with an emphasis on Alzheimer`s disease (AD), the proteins and signalling pathways involved in UPR and it discusses the results and possible utilization of experimental drugs.

The first three chapters of the manuscript are well written and show compelling evidence about the potential importance of the regulation of UPR in dementia.

  1. Line 34: “In England and Wales, the annual mortality from dementia is similar in magnitude to the mortality arising from SARS-CoV-2 in 2020 at nearly 70,000 deaths per year” Authors should also compare the mortality of AD to other chronic diseases. Authors should address the age dependent risk of dementia and AD.

We have now amended the text to make this clearer and added the statement, as requested, that: ‘In England and Wales, the annual mortality from dementia is similar in magnitude to the mortality arising from SARS-CoV-2 in 2020 at nearly 70,000 deaths per year, and greater than that for ischemic heart disease or cerebrovascular disease. The biggest risk factor for developing dementia is advancing age.’

  1. Line 51: “common mechanisms (of AD) are gathering attention as therapeutic targets, due in part to the repeated failure of multi-billion-dollar clinical trials targeting disease-specific proteins, notable Aβ, which has diminished confidence in ever finding dementia treatments” It is not a scientific argument, and these kind of personal opinions and conclusions has no place in scientific reviews. The cited article Mullard A (2019) NLRP3 inhibitors stoke anti-inflammatory ambitions. Nat Rev Drug Discov 18:405–408 is an opinion peace / news article. Stating that there is diminished confidence in finding Ab targeting drugs is simply wrong. The US Food and Drug Administration (FDA) just recently approved a drug targeting Aβ (aducanumab (Aduhelm)). And there are more coming. (https://www.nature.com/articles/d41586-021-03410-9) Authors should stick to the facts without injecting opinions into their paper.

We agree that the focus of this paper is not the debate about the utility or otherwise of agents targeting beta amyloid and we have removed this sentence and the references highlighted.

  1. Line 109-110. Figure 1 shows the activation of UPR “associated with Pathological Protein deposition in Neuro-109 degenerative Disorders”.

Authors need to increase the clarity of how UPR can affect beta-amyloid (Ab) aggregation in AD. The common understanding is the Ab accumulation occurs in the extracellular space and leads to the formation of senile plaques. The common knowledge about Amyloid precursor protein (APP) is that it is a transmembrane protein located in the cell membrane. The authors are right when they state “Most cases of progressive neurodegenerative diseases are characterized by the misfolding of proteins which aggregate into intraneuronal proteinaceous inclusions such as Aβ and tau in Alzheimer’s disease” but they also should discuss that Ab accumulates outside of the cell. Authors should elaborate on the locations of Ab peptide formation (ER, Golgi, cell membrane) and locations of Ab aggregation (ER, Extra cellular space). The following publications (or similar ones) can be cited to back up these claims: LaFerla, F., Green, K. & Oddo, S. Intracellular amyloid-β in Alzheimer's disease. Nat Rev Neurosci 8, 499–509 (2007). https://doi.org/10.1038/nrn2168. Del Prete et al. J Alzheimers Dis. 2017; 55(4): 1549–1570.

We have added the following text to make this clearer and we have added the references suggested ‘Though most of these proteins are intracytoplasmic it is important to note the Aβ, though processed in the endoplasmic reticulum and Golgi, accumulates primarily in extracellular plaques, though intraneuronal accumulation has also been described’.

  1. The main problem with the manuscript is in chapter 5, 6 and 7. In these chapters the authors focus on the drug Trazodone and discuss its potential in the treatment of AD. It is unfortunate that the authors did not exercise the expected and required scientific rigor during this discussion.

These chapters include a lot of assumptions about why Trazodone did not work in clinical trials in the past to treat AD. The main issue here is that the authors avoided to give a clear scientific argument about why previous clinical trials were inconclusive but clearly argue that the conclusion of the trials – Trazodone does not treat AD – is wrong.

We thank the reviewer for the opportunity to address this issue.  We apologise for lack of clarity and scholarliness in the first submission. We have amended the text considerably to clearly reference and explain our reasoning as to why observational studies and short-duration, low dose, clinical trials of trazodone for sleep and agitation do not allow a controlled assessment of trazodone effects on cognition.

We have amended the manuscript to include the following points on pages 6,7 and 8, all appropriately referenced.

-              primarily that the trials have been very short in duration (often just 10-12 weeks in clinical trials)

-              the drug used at likely insufficient doses, given our preclinical data (discussed below).

-              We emphasise that observational evidence is inevitably subject to confounding and is not suited to providing objective evidence of efficacy: such evidence requires a randomised control trial.

-              We have included a new systematic review from Nov 2021 summarising all the clinical studies of trazodone which measured cognition as an outcome.

  1. Line 209 “This study found no positive impact on cognition compared to treatment with other antidepressants. However, the patient group had relatively advanced disease which may have been past the point of possible positive impact and doses were lower than those proposed to impact the UPR (predicted around 200mg [14]), with a mean of 100mg in this study.”

“Relatively advanced disease” is not a scientific term but a lingo what we use in the cafeteria when we have causal chat with peers and colleagues. Authority should disclose a scientific argument citing data and evidence about what disease severity could be treated with Trazodone indicating the disease severity with currently used dementia ratings.

The reviewer is correct. We have modified the text to include the well-established way of describing disease severity in Alzheimer’s disease with currently used dementia ratings is to divide mild moderate and severe disease by MMSE scores, with mild disease equating to scores 20-30, moderate disease 10-20 and severe disease 0-10.  We have amended the text

  1. The insufficient dosage of Trazodone is a very good argument. If previous clinical trials were done with insufficient dose than the outcome of these studies can be deemed inconclusive. Authors should elaborate on what dosage would be sufficient and disclose more relevant details especially because the cited paper (Ref #14 Halliday M, Radford H, Zents KAMM, et al (2017) Repurposed drugs targeting eIF2α-P-mediated translational repression prevent neurodegeneration in mice. Brain 140:1768–1783.) used the following dose: “Mice were intraperitoneally injected once daily with 40 mg/kg trazodone hydrochloride”. That would be a lot of mgs (more than 200) for a 70kg human being. At this point it is unclear why 200mg would be sufficient to treat AD and based on the animal experiment it is unclear what dosage should be sufficient for humans to treat AD. Authors should address this issue and discuss the dosage of Trazodone.

The conversion of dosing in mouse to equivalent human doses was described in the original paper in Brain by Halliday et al. using conversion formulae published in Reagan-Shaw et al., 2008, cited in that paper. 

The text below is taken from Halliday et al, for rapid reference for the Reviewer’s convenience. However, we provide hyperlinks to both Halliday et al and Reagan-Shaw et al.

‘We next tested the therapeutic effects of trazodone and DBM in vivo. For clinical relevance, we used a daily dose of 40mg/kg of trazodone delivered intraperitoneally in mice, equivalent to 194 mg/day in humans. Patients usually receive 150–375 mg trazodone per day. We used a standard conversion formula for dose translation to calculate the corresponding mouse dose (Reagan-Shaw et al., 2008)’

Thus, we did not use a simple conversation based on body weight, but one using body surface area normalisation according to published protocol (Reagan-Shaw et al., 2008).  We have amended the text on page 6 to make this clearer

  1. Similar problems arise at line 225-230. Authors disclose opinions and speculations without providing any evidence or facts. Authors should disclose evidence so their opinion can be recognized as a scientific message.

We thank the reviewer for the opportunity to amend this.  We have added text on page 7(see below) and the relevant references to support the suggestion that trazodone is an uncommon choice of antidepressant and, when used in dementia, is commonly done so to treat agitation which is a marker of advanced disease. 

New text p7: “For example, a recent study of 465,628 patients aged over 50 prescribed antidepressants, only 4,596 were prescribed trazodone as first line treatment (<1%) [70]. In another study of 2199 patients, the most common reason for starting trazodone was agitation, not depression [71].”

  1. Line 224-226 “Thus, these studies do not show a clear effect of trazodone use on dementia incidence or cognition – with one exception [53], however, observational studies can only ever show associations and have many confounding variables.” Authors should provide more details about that “one exception”. Author should discuss this study and compare it to the other studies which had negative outcome. Is there a difference between them what would help to design future studies to evaluate Trazodone?

This is a key point, and we thank the reviewer.  We have modified the text to reflect the fact that the exception measured MMSE and its decline in patients with mild AD (MMSE >20) and MCI and over the course of 4 years. We have amended the text on page 6.

  1. Line 227-230 “For example, trazodone is an unusual choice of antidepressant for older people, and it may be prescribed preferentially in those with symptoms suggestive of incipient dementia (for example sleep disturbance) or in established dementia for those with behavioural symptoms which are associated with more advanced disease, both of which would mask any benefits of the drug.” Author must cite relevant literature to back up this claim. –

We have added text and the relevant references to support the suggestion that trazodone is an uncommon choice of antidepressant and when used in dementia is commonly done so to treat agitation which is a marker of advanced disease on page 7. (See point 7 above also).

  1. Line 231 “In summary” Authors disclose opinions and speculations without providing any evidence or facts. Authors should disclose evidence so their opinion can be recognized as a scientific message. What are the specific shortcomings of the cited studies and would be the needed tests to prove the efficacy of Trazodone.

We have amended the text as outlined above to highlight the main shortcomings of existing data – that it is observational or trials have been for short periods of time, at low doses and in in late disease.  We have taken the reviewer’s advice and included a description of what patient group/trial design might improve detection of efficacy of trazodone, notably experimental and randomised control trial approaches in early disease, at sufficient doses and for appropriate duration.  We also indicate key mechanistic questions that should be answered in patients in the final paragraph, p8.

  1. Line 266-268 “Taken together…” Authors, again disclose opinions and speculations without providing any evidence or facts. Authors should disclose evidence so their opinion can be recognized as a scientific message. What would be a long enough treatment and why?

We have amended the text on p8 to explicitly address these concerns and this now reads:

New text p8: ‘Most published trials of trazodone in dementia are for duration of treatment measured in weeks, and none reach consensus statements for clinical trials of disease modifying drugs in dementia which stipulate a minimum trial duration of 18 months [94]’.

  1. My personal opinion is that I agree with the authors that Trazodone needs a closer look, and it has the potential in the treatment of AD. But my opinion does not matter because science is not about opinions but verified and well-established facts. Every hypothesis must be built on facts. I encourage the authors to provide sufficient evidence and facts to the claims of their review article.

We are very grateful to the Reviewer for the opportunity to clarify our arguments, which we have now fully referenced and included discussion of observational studies, clinical trials and a systematic review to support the views expressed.  We have amended the text significantly to address specific concerns and make the article clearer and more tightly referenced and to present our hypothesis clearly in the context of the current landscape.  We hope the Reviewer finds this is now much more compellingly argued.

Reviewer 2 Report

In the manuscript 'Targeting the Unfolded Protein Response as a Disease Modifying Pathway in Dementia,' the authors have described the role of UPR, and its over-activation in pathogenicity in dementia. The use of UPR-modifying agents as therapeutic targets in dementia and future potential benefits are discussed in great detail

The article is clearly written, and the authors have added beautiful figures. I congratulate the authors for this nice paper.

Author Response

Reviewer 2

In the manuscript 'Targeting the Unfolded Protein Response as a Disease Modifying Pathway in Dementia,' the authors have described the role of UPR, and its over-activation in pathogenicity in dementia. The use of UPR-modifying agents as therapeutic targets in dementia and future potential benefits are discussed in great detail

The article is clearly written, and the authors have added beautiful figures. I congratulate the authors for this nice paper.

We thank Reviewer 2 for their consideration and generous comments.

Reviewer 3 Report

In general, I enjoyed reading your brief and comprehensive paper. However, I have a few remarks. When describing possible potential drugs to treat UPR you did not mention recombinant human Hsp70 which was extensively studied in this respect in various countries in numerous in vitro and rodent models of neurodegeneration (e.g. by the Asea group in the USA and by Evgen'ev's group in Russia). It will be also appropriate to mention the first FDA-approved drug (Tacrine) to use in AD patients. Besides, I recommend decreasing a section of your MS where you describe human studies of  Trezodone that provided rather conflicting results.

Author Response

Reviewer 3

In general, I enjoyed reading your brief and comprehensive paper.

  1. However, I have a few remarks. When describing possible potential drugs to treat UPR you did not mention recombinant human Hsp70 which was extensively studied in this respect in various countries in numerous in vitro and rodent models of neurodegeneration (e.g. by the Asea group in the USA and by Evgen'ev's group in Russia).

We agree that recombinant Hsp70 and other chaperones are an important therapeutic approach in neurodegeneration with an extensive literature.  However, it has not, to our knowledge, been shown to directly impact on the unfolded protein response, which is the focus of this review.  As such we feel that recombinant Hsp70 is beyond the scope of this review.

  1. It will be also appropriate to mention the first FDA-approved drug (Tacrine) to use in AD patients.

We have amended the text on p1  as follows and added the appropriate references:

New text p1 “The current drug treatments available, such as cholinesterase inhibitors developed after initial trials of tacrine and the NMDA receptor antagonist, memantine, may ameliorate the symptoms of disease, but they do not delay its progression”

  1. Besides, I recommend decreasing a section of your MS where you describe human studies of Trazodone that provided rather conflicting results.

We have re-written and edited this section to take in to account these comments and those of the other reviewers in an effort to increase clarity. Amended text has been highlighted where we seek to reduce the length of this section by making our arguments clearer.

Round 2

Reviewer 1 Report

Dear Authors,

Thank you for answering all of my questions. 
I have no further comments on your manuscript.